# Fine Particulate Matter Perturbs the Pulmonary Microbiota in Broiler Chickens

**DOI:** 10.3390/ani13182862

**Published:** 2023-09-08

**Authors:** Ying Zhou, Bin Xu, Linyi Wang, Chaoshuai Zhang, Shaoyu Li

**Affiliations:** Institute of Animal Husbandry and Veterinary Science, Henan Academy of Agricultural Sciences, Zhengzhou 450002, China; zhouying0121@126.com (Y.Z.); xubin9941@126.com (B.X.); wlyzmx@126.com (L.W.); zcs1109584786@163.com (C.Z.)

**Keywords:** particulate matter, pulmonary microbiota, inflammation, histological, broiler

## Abstract

**Simple Summary:**

Particulate matter (PM) is the most important hazardous pollutant and seriously affects the respiratory tract health of both animals and humans. Nowadays, the development of intensive animal husbandry not only increases PM pollution in the atmospheric environment but also harms the health of animals and ranch workers. The concentration of PM in poultry houses is higher than that for other animals, such as in pig and cow houses. However, there are few studies on the effect of fine particulate matter on pulmonary microbiota in poultry. This study aims to explore the effect of fine particulate matter on pulmonary microbiota in broilers.

**Abstract:**

(1) Fine particulate matter (PM_2.5_) seriously affects the respiratory tract health of both animals and humans. Growing evidence indicates that the pulmonary microbiota is involved in the development of respiratory tract health; however, there is still much that is unknown about the specific changes of pulmonary microbiota caused by PM_2.5_ in broilers. (2) In this experiment, a total of 48 broilers were randomly divided into a control group and PM-exposure group. The experiment lasted for 21 days. Microbiota, inflammation biomarkers, and histological markers in the lungs were determined. (3) On the last day of the experiment, PM significantly disrupted the structure of lung tissue and induced chronic pulmonary inflammation by increasing IL-6, TNFα, and IFNγ expression and decreasing IL-10 expression. PM exposure significantly altered the α and β diversity of pulmonary microbiota. At the phylum level, PM exposure significantly decreased the *Firmicutes* abundance and increased the abundance of *Actinobacteria* and *Proteobacteria*. At the genus level, PM exposure significantly increased the abundance of *Rhodococcus*, *Achromobacter*, *Pseudomonas*, and *Ochrobactrum*. We also observed positive associations of the above altered genera with lung TNFα and IFNγ expression. (4) The results suggest that PM perturbs the pulmonary microbiota and induces chronic inflammation, and the pulmonary microbiota possibly contributes to the development of lung inflammation.

## 1. Introduction

Air pollution is well known to be a potential danger to public health throughout the world [1]. Among pollutants, haze poses potential health issues for most cities in China [2]. Fine particulate matter (PM_2.5_) is the main component of haze [3,4]. Nowadays, the development of intensive animal husbandry not only increases PM_2.5_ pollution in the atmospheric environment but also harms the health of animals and ranch workers [5]. According to the specific statistics of poultry houses equipped with mechanical ventilation systems, the PM_2.5_ concentrations ranged from 40 to 2530 µg/m^3^ in general [6,7,8,9] and were 67~1480 µg/m^3^ in the spring, 67~1370 µg/m^3^ in the summer, 1230~1920 µg/m^3^ in the fall, and 40~2530 µg/m^3^ in the winter [6,8,10]. Moreover, the concentration of PM in poultry houses is higher than that for other animals, such as in pig and cow houses [5]. Unlike the lungs of mammals, broiler lungs contain multiple bronchi and airbags. The unique structure keeps the lungs of broilers in a semi-open state, making them more susceptible to PM damage [11]. And chronic high PM concentrations exposure can make broilers susceptible to respiratory diseases, including chronic bronchitis, asthma, and dust poisoning syndrome [12,13], which result in a significant annual loss from respiratory disease within the poultry industry [14]. Thus, it is necessary to investigate lung injury under PM exposure in broilers.

Although the mechanism of lung injury caused by PM_2.5_ has not been fully elucidated, increasing evidence suggests that the imbalance of pulmonary microbiota is closely related to lung injury. The pulmonary microbiota is dynamically balanced in terms of health [15], changed in the condition of disease [16], related to the changes in alveolar immunity [15], and also is a predictive indicator of lung diseases [17,18]. Some studies have revealed characteristic changes in pulmonary microbiota associated with disease phenotype, such as in pulmonary fibrosis [19], COPD [20], asthma [21], pneumonia [22], and other chronic respiratory diseases [23]. Previously published studies have documented that PM_2.5_ caused pulmonary microbiota alterations in humans [24], mice [25,26], and rats [27]. Thus, considering the unique lung structure of poultry, the specific changes of the pulmonary microbiota induced by PM_2.5_ are also essential to elucidate in broilers.

To the authors’ knowledge, there are few studies on the effect of fine particulate matter on the pulmonary microbiota in poultry. Therefore, the present study aimed to investigate the effects of PM_2.5_ on the pulmonary microbiota. Further, we also evaluated the effects of PM_2.5_ on the pulmonary histological changes and inflammation biomarkers in broilers. This study will investigate the potential relationship between the pulmonary microbiota and pulmonary injury in broilers under PM_2.5_ exposure. The present study fills the gap in the research concerning the impact of fine particulate matter on the pulmonary microbiota in poultry.

## 2. Materials and Methods

### 2.1. PM Preparation

PM (NIST, 1649b) with a particle size distribution ranging from 0.2 µm to 2.5 µm was obtained from the National Institute of Standards and Technology. The standard PM_2.5_ was composed of 20 polycyclic aromatic hydrocarbons (PAHs), 8 nitro-substituted PAHs (nitro-PAHs), 13 polychlorinated biphenyl (PCB) congeners, 4 chlorinated pesticides, and 3 inorganic constituents. All of the constituents for which certified, reference, and information values are provided in SRM 1649b are naturally present in the particulate material.

### 2.2. Animal Experiment

In the study, the animal experiment was ratified by the Ethics and Clinical Research Committee of Henan academy of agricultural sciences institute of animal sciences. One-day-old Arbor Acres (AA) broilers obtained from the commercial broiler farm were reared in cages within the animal husbandry laboratory. The housing conditions were set at 23 ± 1 °C for temperature, 60 ± 5% for relative humidity, and 24 h for light per day. A complete diet was given to the broilers from day 1 to day 42, meeting the NRC (1994) standards.

Before the exposure experiment, 14-day-old AA broilers were adapted to the chambers for 1 week. Then, 21-day-old AA broilers with average body weight were randomly divided into 2 groups, including the control group (CON) and PM_2.5_-exposure group (PM), with 24 birds in each group. The broilers in the control group were exposed to normal saline, while the broilers in the PM_2.5_ group were exposed to PM_2.5_ with a concentration of 1000 µg/m^3^. Broilers were kept in cages in a room during non-exposure time and transferred to two identical independent exposure boxes during exposure time to ensure consistency in other environmental factors such as temperature, humidity, etc., except for PM_2.5_ concentration. The experiment period lasted 21 d.

### 2.3. PM Exposure Treatment

In the present study, we used the method of a whole-body exposure to simulate the real exposure status. And the exposure dose calculation formula actually used in this study is as follows: Amount of single inhalation of PM_2.5_ = PM_2.5_ mass concentration × working efficiency of physicochemical device × respiratory capacity of broiler × exposure time. Based on the PM_2.5_ concentration and exposure time set up in this experiment, the final dose of PM_2.5_ to each chicken is as follows: 600 mL × 1000 µg/m^3^ × 120 ÷ 10^6^ × 100/6.67 = 1079 µg. In this equation, 600 means the ventilation volume per minute of each chicken, 6.67 means the settling rate of PM_2.5_ in the lung of broilers, and 120 means 2 h exposure time. Finally, we took the corresponding dose of PM_2.5_ for the PM group according to the number of broilers.

Before daily exposure, the broilers were transferred into the two same independent exposure boxes, which were 1.2 m in length, 1.0 m in width, and 0.5 m in height. Broilers were exposed to PM or normal saline for a 21-day exposure of 2 h/d from 8:00 to 10:00 using the NSF-6A model liquid aerosol generation system (Shanghai TOW, Shanghai, China), which can continuously and stably generate aerosol aerosols for solutions and maintain the particle size distribution generally at 0 within 2–3 µm. The specific exposure method was as follows: one end of the hose was connected to the exposure box, and the other end of the hose was connected to the liquid aerosol generation system. The liquid aerosol generation system firstly introduced the PM or normal saline through the injection port, and then, the PM or normal saline were sprayed into the exposure box through the hose under the pump power action. The atomizer of NSF-6A controlled the injection speed of PM or normal saline. PM concentrations were also monitored in real time by the IDG100-TSP monitor, which was placed in the exposure box. After exposure every day, the broilers were returned to the cages. The exposure boxes were washed with clean water and also irradiated with an ultraviolet lamp for about 4 h.

### 2.4. Sample Collection

At day 42 of the experiment, six broilers with average body weight were euthanized by cervical dislocation; lung tissues were taken for further molecular mechanism analysis.

One small part in the middle of left lung tissues was placed in 4% paraformaldehyde. Another small part in the middle of right lung tissues was taken and quickly frozen with liquid nitrogen and then kept at −80 °C for further analysis of the molecular mechanisms by using real-time quantitative (RT-PCR) method. The microbiome in lung tissues was analyzed by using 16 S rRNA sequencing.

### 2.5. Real-Time RT-PCR

Lung inflammatory injury markers including IL-1β, IL-6, IL-10, IFN-γ, TNF-α, and mRNA expression were measured by the RT-PCR. The designed primer sequences for each gene are shown in Table 1, which refers to our previous study [28]. The specific method was as follows: about 800 mg of the lung tissues were taken to prepare the RNA samples using the tissue RNA rapid extraction kit (Imagene). The concentration of extracted RNA was measured by the Nanodeop lite (Thermo Scientific, Waltham, MA, USA). Then, cDNA was synthesized using All-in-One First-Strand Synthesis Master Mix (Kemix, Bejing, China). RT-qPCR was used 2 × SYBR Green qPCR Premix (Kemix, Bejing, China) with a two-step real-time PCR system on the LightCycler 96 system (Roche, Basel, Switzerland). The relative gene expression was calculated by the 2^−ΔΔCT^ method.

### 2.6. Analysis of Histological

Hematoxylin and eosin (H&E) was used to analyze the changes of lung structure. The specific procedures were as follows: a small part of the lung tissue was taken and fixed in 4% paraformaldehyde. First, the lung tissue was dehydrated with gradient alcohol, then made transparent and soaked in wax, embedded, sliced, roasted, and dewaxed. Finally, it was stained with hematoxylin eosin staining solution, air-dried, and sealed with neutral gum. Finally, it was examined under microscope.

### 2.7. S rRNA Sequencing of Pulmonary Microbiota and Bioinformatics Analysis

16 S rRNA sequencing was used to measure the alterations in the pulmonary microbiota, and the specific steps were as follows in Table 2.

### 2.8. Statistical Analysis

The above data regarding IL-10, IL-1β, IFN-γ, IL-6, and TNFα were analyzed by Student’s *t*-test using SPASS 26.0 software. Mean ± SE was used to present the data. *p* < 0.05 was set as “significance”. GraphPad Prism 5.0 was used for plotting the above data.

## 3. Results

### 3.1. Pulmonary Injury

Pulmonary injury is mainly evaluated from two aspects: histological changes and markers of inflammatory factors. The histological changes are shown in Figure 1; in the PM group, various sizes of alveoli can be observed, and at the same time, the alveoli disappear significantly, with more alveolar damage (black). There was a large amount of inflammatory infiltration in the parenchymal and epithelial areas, and some vacuoles appeared (red). In addition, PM exposure significantly increased the expression of IL-6, TNFα, and IFNγ (*p* < 0.05) and significantly decreased the IL-10 expression (*p* < 0.05) in the lung.

### 3.2. Rank–Abundance Curve

Rank–abundance is mainly used to evaluate the saturation of a sample. As shown in Figure 2, it can be seen that the curve based on the Shannon index gradually flattens out, which means that the number of sequencing units are large enough, and the number of OTU species does not increase with the increase of sequencing quantity, suggesting that the tested samples already contain the species information of the vast majority of microorganisms.

### 3.3. Alpha Diversity

Alpha diversity can reflect the species richness and diversity of samples, which is usually measured by indicators such as Chao, Ace, Shannon, and Simpson. Among them, the Chao and Ace indexes are used to measure the species richness, that is, the number of species; Shannon and Simpson indexes are used to measure species diversity. As shown in Figure 3, PM exposure significantly increased the Chao index and decreased the Simpson index, indicating that PM induced the increase of species richness and diversity.

### 3.4. Beta Diversity

The beta diversity analysis compares the similarity of each sample in species diversity. In this experiment, samples were analyzed based on principal coordinate analysis (PCoA) and unweighted_unifrac analysis. The PCoA method mainly separates and classifies multiple samples, which can more deeply reflect the differences between species in the samples. The closer the samples are on the coordinate map, the greater the similarity. Based on the PCoA analysis results, it can be seen from the figure that the contribution values of the first and second principal components in principal coordinate analysis are 31.37% and 21.43%, respectively. Further inter-group difference analyses were conducted using the Anosim method for principal coordinate analysis, and the results showed that there is significant difference between CON and PM groups (Figure 4, R = 0.7593, *p* = 0.023), indicating there is a significant separation of bacterial communities between the control group and the PM group, suggesting that PM treatment has a significant impact on the composition of the pulmonary microbiota.

### 3.5. Pulmonary Microbiota Composition at the Phylum and Genus Levels

QIIME software (1.9.1) was used to generate the species-richness table and draw a species distribution histogram at both the phylum and genus levels, where the color represents species, and the color block length represents the proportion of species in the relative abundance (Figure 5). At the phylum level, the top phylum are *Firmicutes*, *Actinobacteriota*, *Proteobacteria*, and *Bacteroidota*. And the relative abundance ratios of phylum level in the CON group are 83.89%, 1.29%, 2.19%, and 8.29%, respectively; and the relative abundance ratios of phylum level in the PM group are 28.04%, 56.23%, 10.12%, and 1.19%, respectively. As shown in Figure 5, *Firmicutes* abundance in the PM group decreased by 66.57%, while *Proteobacteria* increased by 78.36% in the PM group.

In the genus level, the top genus are *Staphylococcus*, *Rhodococcus*, *unclassified_k_norank_d_Bacteria*, *unclassified_f_Lachnospiraceae*, *Achromobacter*, *Romboutsia*, *Lactobacillus*, *Faecalibacterium*, *Intestinimonas*, *Pseudomonas*, *Ruminococcus_torques_group*, *Parabacteroides*, *Blautia*, *Bacteroides*, *norank_f_norank_o_Clostridia_UCG-014*, *norank_f_norank_o_Clostridia_vadinBB60_group*, *norank_f_Flavobacteriaceae*, *UCG-005*, and *Alistipes*.

### 3.6. Test of Microbiota Composition between the CON and PM Groups at the Phylum and Genus Levels

To further investigate the alterations in microbiota composition between the CON and PM groups, Student’s *t*-test was used to analyze the difference. As shown in Figure 6, at the phylum level, PM exposure significantly decreased the *Firmicutes* abundance and significantly increased the abundance of *Actinobacteria* and *Proteobacteria* (*p* < 0.05). In addition, at the genus level, PM exposure significantly increased the abundance of *Rhodococcus*, *Achromobacter*, *Pseudomonas*, and *Ochrobactrum* (*p* < 0.05), while PM exposure significantly decreased the *Ruminococcus_gauvreauii_group*, *Sphingomonas*, *Anaerofilum*, *Cellulosilyticum*, *Actinoplanes*, *norank_f_JG30-KF-CM45*, *norank_f_Ilumatobacteraceae*, *Subgroup_10*, *Nordella*, *norank_f_Vicinamibacteraceae*, and *norank_f_Xanthobacteraceae* (*p* < 0.05).

### 3.7. Correlation between Pulmonary Microbiota and Inflammation under PM Exposure

As shown in Figure 7, correlation analysis was used to analyze the correlations between the top 50 most abundant genera and IL-10, IL-6, TNFα, and IFNγ expressions. The results showed that Ochrobactrum (r = 0.94, *p* = 0.0048; r = 0.93, *p* = 0.00767), Rhodococcus (r = 0.94, *p* = 0.0048; r = 0.93, *p* = 0.00767), Delftia (r = 0.93, *p* = 0.00767; r = 0.94, *p* = 0.00509), Pseudomonas (r = 0.88, *p* = 0.01885; r = 0.93, *p* = 0.00767), and Acinetobacter (r = 0.94, *p* = 0.01885; r = 0.93, *p* = 0.00767) were significantly correlated with TNFα and IFNγ expression (*p* < 0.05). Oscillibacter (r = 0.88, *p* = 0.01982), Shuttleworthia (r = 0.85, *p* = 0.03411), and Turcibacter (r = 0.81, *p* = 0.04986) was significantly correlated with IL-10 expression (*p* < 0.05).

## 4. Discussion

This study mainly investigated the effects of PM on the pulmonary microbiota composition, pulmonary cytokines and histological changes, as well as the potential relationship between the pulmonary microbiota and injury under PM_2.5_ exposure. Our research showed that PM exposure changed α and β diversity, altered the pulmonary microbiota composition, and also induced pulmonary inflammation. In addition, correlation analysis indicated pulmonary microbiota were positively correlated with pulmonary inflammation under PM exposure.

Increasing evidence indicated that the interactions between the pulmonary microbiota and host have a significant role in maintaining pulmonary health [29,30,31,32,33,34]. Studies have also illustrated the changes in pulmonary microbiota composition, such as that which occurs in COPD [20], asthma [21], and pneumonia [22]. In the present study, we characterized the alterations in the pulmonary microbiota of broilers under PM exposure. First, through four indexes of α diversity (Ace, Chao, Shannon, and Simpson), we elevated the richness and diversity, and the results indicated that PM increased the species richness and diversity. Also, PM exposure changed the structure of the lung microbiota. Importantly, PM significantly altered the lung microbiota composition. At the phylum level, PM exposure significantly decreased the *Firmicutes* abundance and significantly increased the abundance of *Actinobacteria* and *Proteobacteria*. The literature suggests that a decrease in the abundance of *Firmicutes*-associated microbiota in the respiratory tract is associated with the development of pneumonia [22]. A study using both primary lung tissue samples and a validation cohort from the Cancer Genome Map (TCGA) showed an overall increase in *Proteobacteria* in the lung cancer microbiome, while the abundance of *Acidophilic Bacillus acidovorax* (*Proteobacteria*) increased in squamous cell carcinoma with TP53 mutations in smokers, indicating the interaction between microbiome genes and microbiota exposure [35]. The above results indicated that PM exposure altered the diversity and structure of lung microbiota and also induced pulmonary-disease-associated lung microbiota.

The main biological mechanism of lung injury caused by PM is pulmonary inflammation, which is also the cause of the development and worsening of lung disease caused by PM [36]. The alteration in cytokines concentration was closely related to respiratory disease occurrence. TNFα is an inflammatory factor that mainly induces airway hyperresponsiveness in animals and humans, which is significantly associated with damage to airway epithelial tissue, activation and chemotaxis of eosinophils, and release of basic proteins. Research has found that PM_2.5_ increased TNFα expression by regulating NF-kB, resulting in airway inflammatory response and tissue damage [37]. It was also reported that IFNγ is a type II that possesses various biological activities, including antivirus effect and antitumor effect. IL-6 is a multifunctional cytokine that can regulate immune response and the activation, growth, and differentiation of T cells related to inflammatory response and can promote T-cell-mediated inflammatory response. In addition, it has been proven that during lung injury caused by viral infection, macrophages and epithelial cells release IL-6, which interacts with TNFα synergistic effects and exerts biological effects [38]. Previous studies have found that PM exposure caused an increase in IL-1β, IL-18, and inflammatory cell counts in the lung tissue of mice [39], and PM also induced pulmonary inflammation in human bronchial epithelial cells [40]. In line with the previous study, our results also revealed that PM increased the expression of IL-6, TNFα, and IFNγ and significantly decreased the IL-10 expression in the lung of broilers, which indicates PM exposure induced chronic pulmonary inflammation in broilers. Our results also showed that PM induced results of injury, namely inflammatory infiltrates and hyperemia, indicating that the lung barrier was damaged after exposure. This injury causes cytokines in the lung to enter the bloodstream or vice versa, thereby causing uncontrollable inflammation through multiple signal transduction pathways.

In recent years, research has shown that the lung microbiota also contributes to pulmonary inflammation [18,25,41,42]. In the present study, *Ochrobactrum*, *Rhodococcus*, *Delftia*, *Pseudomonas*, and *Acinetobacter* were significantly correlated with TNFα and IFNγ expression. Moreover, at the genus level, PM exposure significantly increased the abundance of *Rhodococcus*, *Achromobacter*, *Pseudomonas*, and *Ochrobactrum*. Thus, the alterations in the pulmonary microbiota may also contribute to lung inflammation under PM exposure.

Specifically, the *Rhodococcus* genus was discovered in 1977 and includes mycobacteria-like organisms, of which *R. equi* is associated with pulmonary masses/infiltrations/abscesses/cavities [43,44]; these pulmonary abscesses extend to the ipsilateral mediastinum and supraclavicular fossa [45], leading to ipsilateral/unilateral pleural effusions (even with mediastinal (fibrosing pleurisy)) [44,46], bacteremia [47], bilateral maxillary sinusitis, bilateral necrotic pneumonia, bronchogenic fistula [47], and acute respiratory distress/shock/death and other diseases [48]. Inhaling contaminated aerosols seems to be the main form of transmission of *Rhodococcus*, with most cases involving the lungs [49]. *Achromobacter* is a ubiquitous environmental organism that can also become pathogenic pathogens in some situations, such as cystic fibrosis, renal failure, and immunodeficiency [50]. Another study also showed that cystic fibrosis patients were the most frequently colonized subjects by *Achromobacter* spp., which can cause continuous airway infections [51]. *Pseudomonas* is the culprit of various infections, especially those involving the air passage [52,53,54], which may produce severe pneumonia in immunocompromised individuals [55], cause chronic obstructive pulmonary disease [56], induce bronchiectasis pathogenesis and contribute to airway inflammation and epithelial damage in bronchiectasis [57], and trigger allergic inflammation [58] or lung inflammation [59]. The increase of genus *Ochrobactrum* is related to underlying disorder or disease that increases individual susceptibility to infection, and *Ochrobactrum* could act as an opportunistic pathogen in immunocompromised individuals [60]. In addition, *Ochrobactrum* may also be associated with some co-morbidities such as pneumonia, hypertension, and diabetes mellitus [61]. Growing evidence found that *Ochrobactrum* could infect immunocompetent hosts with, for example, endocarditis and septicemia [62,63]. Thus, the above results further indicate that PM exposure increases the lung disease and inflammation associated with pulmonary microbiota, which may be due to the host susceptibility induced by PM exposure.

Additionally, PM exposure significantly decreased *Ruminococcus_gauvreauii_group*, *Sphingomonas*, *Anaerofilum*, *Cellulosilyticum*, *Actinoplanes*, *norank_f_JG30-KF-CM45*, *norank_f_Ilumatobacteraceae*, *Subgroup_10*, *Nordella*, *norank_f_Vicinamibacteraceae*, and *norank_f_Xanthobacteraceae*. Among them, some strains of *Ruminococcus_gauvreauii_group* can degrade mucin to produce propionic acid, providing energy to the host and promoting their own colonization [64]. The lower abundance of *Sphingomonas* was possibly correlated with the formation of mucus plugs in children with *Mycoplasma pneumoniae pneumonia* [65]. *Cellulosilyticum* is the cellulose-degrading genus, which may be related to host health and fat deposition [66]. Due to the much lower abundance of the remaining microbiota, we will not elaborate on them here. Thus, the alterations in the above microbiota indicate that PM exposure decreases the mucus- and energy-associated microbiota in the lung.

## 5. Conclusions

In conclusion, the results indicate that PM exposure increases the pulmonary microbiota associated with respiratory tract disease and inflammation and also disrupts the pulmonary histological structure and induces chronic pulmonary inflammation. Moreover, the alterations in the pulmonary microbiota contribute to the pulmonary inflammation by increasing the levels of TNFα and IFNγ. There findings will help improve our understanding of the possible mechanism by which particulate matter affects the respiratory tract health of broiler chickens. To address a current limitation, we should also set more levels of PM_2.5_ to investigate the relationship between the pulmonary microbiota and pulmonary injury. The future research will focus on the specific mechanism of the pulmonary microbiota in lung injury induced by PM_2.5_.

## Figures and Tables

**Figure 1 animals-13-02862-f001:**
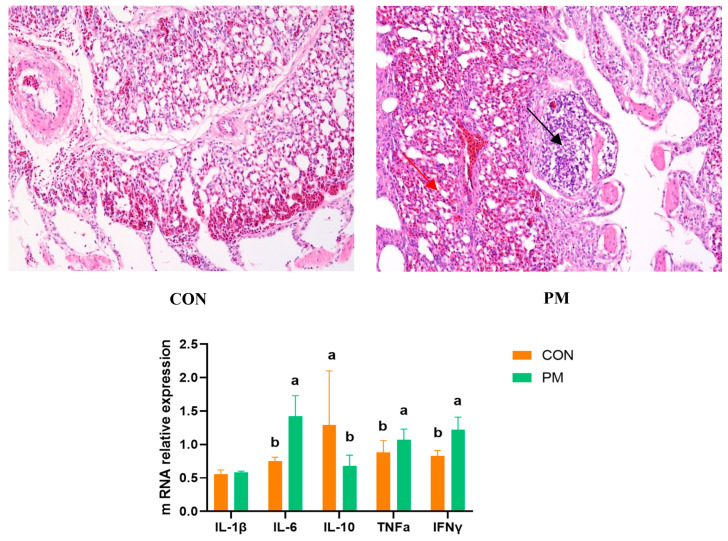
The effect of PM on the lung histological changes and expression of genes associated with inflammation. a,b: means differ significantly (*p* < 0.05).

**Figure 2 animals-13-02862-f002:**
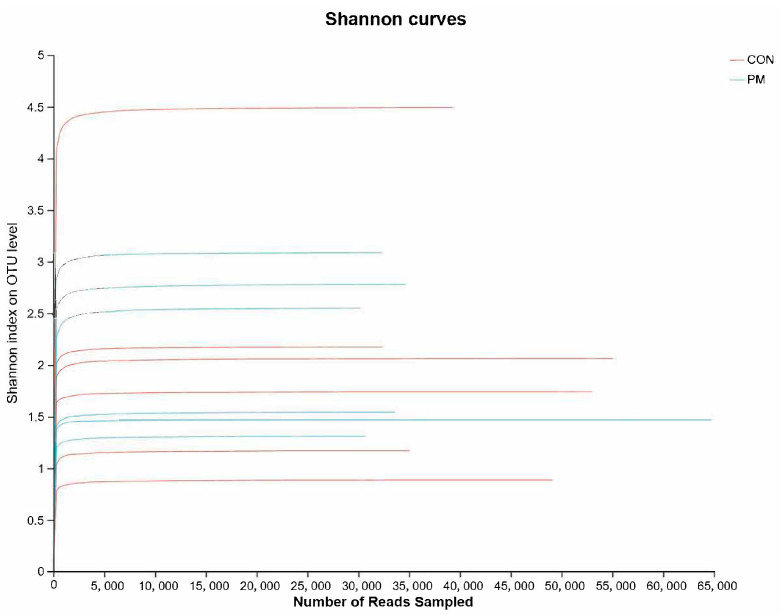
The effect of PM on the rank–abundance curve based on the Shannon index.

**Figure 3 animals-13-02862-f003:**
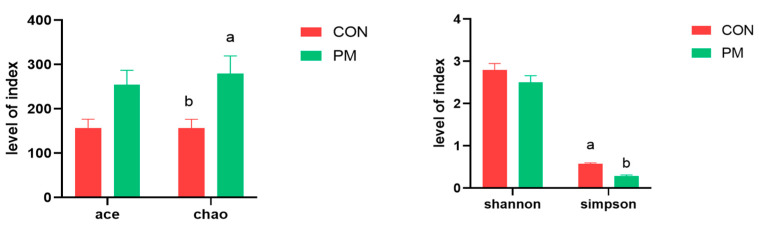
The effect of PM on the index of alpha diversity. a,b: means differ significantly (*p* < 0.05).

**Figure 4 animals-13-02862-f004:**
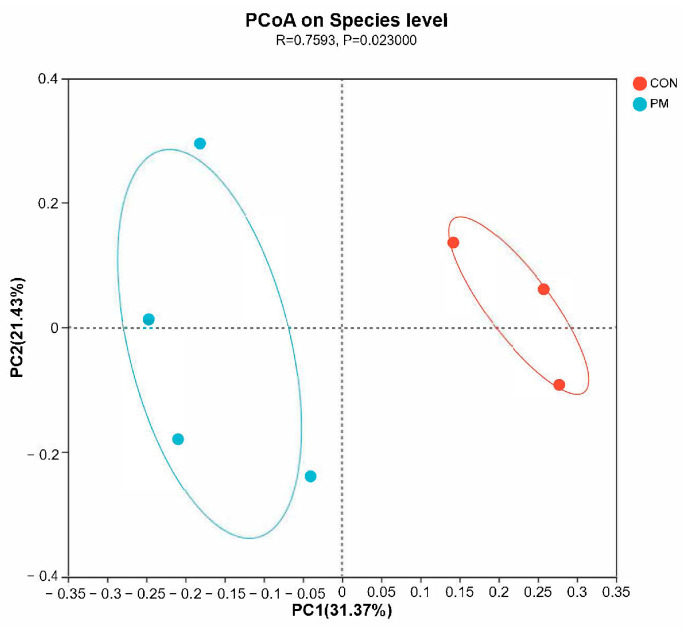
The effect of PM on the beta diversity.

**Figure 5 animals-13-02862-f005:**
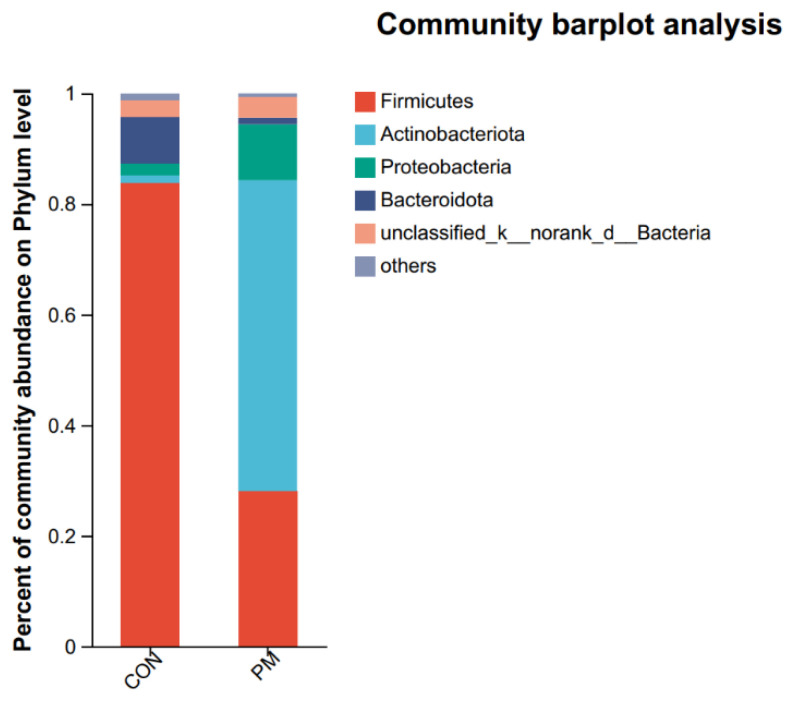
The effect of PM on the pulmonary composition at the phylum and genus levels.

**Figure 6 animals-13-02862-f006:**
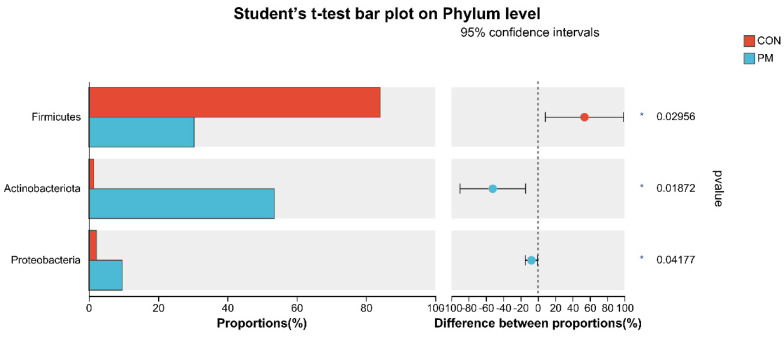
The test of pulmonary microbiota composition at the phylum and genus levels. * means differ significantly (*p* < 0.05). ** means differ highly significantly (*p* < 0.01). *** means differ highly significantly (*p* < 0.001).

**Figure 7 animals-13-02862-f007:**
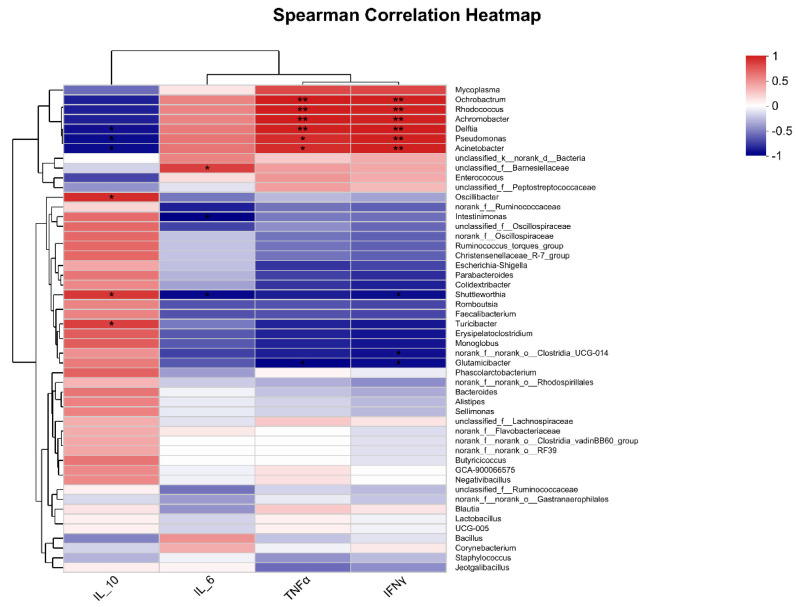
Correlation analysis between pulmonary microbiota and pulmonary cytokines expression. * means a significantly positive correlation (*p* < 0.05) in red cells or a significantly negative correlation (*p* < 0.05) in blue cells. ** means a highly significantly positive correlation (*p* < 0.01) in red cells or a highly significantly negative correlation (*p* < 0.01) in blue cells.

**Table 1 animals-13-02862-t001:** The primers used in the study.

Target Gene	Primer Sequence (5′ to 3′)	Length	Login ID
GAPDH	F:TGAAAGTCGGAGTCAACGGAT	230 bp	NM_204305.1
	R:ACGCTCCTGGAAGATAGTGAT		
IL-1β	F:AGAAGAAGCCTCGCCTGGAT	131 bp	NM_204524.1
	R:CCTCCGCAGCAGTTTGGT		
IFN-γ	F:AGTCAAAGCCGCACATCAAACAC	133 bp	NM_205149.1
	R:CGCTGGATTCTCAAGTCGTTCATC		
TNF-α	F:GGACAGCCTATGCCAACAAG	168 bp	NM_204267.1
	R:ACACGACAGCCAAGTCAACG		
IL-6	F:CCTCCTCGCCAATCTGAAGTCA	210 bp	NM_204628.1
	R:AACGGAACAACACTGCCATCTG		
IL-10	F:ATCCAACTGCTCAGCTCTGAACTG	101 bp	NM_001004414.2
	R:GGCAGGACCTCATCTGTGTAGAAG		

**Table 2 animals-13-02862-t002:** The specific steps of the 16 S rRNA sequencing.

Steps	Kit or Instrument	Measuring Methods
1. DNA extraction	E.Z.N.A.^®^ soil DNA Kit (Omega Bio-tek, Norcross, GA, USA),	Referring to kit’s instructions
2. DNA detection	determined by 1.0% agarose gel electrophoresis and a NanoDrop^®^ ND-2000 spectrophotometer (Thermo Scientific Inc., Waltham, MA, USA)	According to instrument’s instructions
3. PCR amplification	ABI GeneAmp^®^ 9700 PCR thermocycler (ABI, Los Angeles, CA, USA)	V3-V4 of the bacterial 16S rRNA gene: primer pairs 338F (5′-ACTCCTACGGGAGGCAGCAG-3′) and 806R (5′-GGACTACHVGGGTWTCTAAT-3′); the PCR reaction mixture and PCR amplification cycling conditions were based on our previous research (Zhou et al., 2022 [28])
4. Purification and quantification of PCR products		According to kit’s instructions
5. Illumina MiSeq sequence		Referring to the previous research (Zhou et al., 2022 [28])
6. Bioinformatic analysis	Majorbio Cloud platform (https://cloud.majorbio.com, accessed on 24 November 2022)	Alpha diversity indices, principal coordinate analysis (PCoA), spearman’s correlation, etc.

## Data Availability

Not applicable.

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
