# Peer review of "Fine Particulate Matter Perturbs the Pulmonary Microbiota in Broiler Chickens"

_animals, 2023, doi:10.3390/ani13182862_

Round 1
Reviewer 1 Report
The manuscript provides a comprehensive overview of air pollution, specifically on delicate particulate matter (PM2.5) and its impact on broilers’ health. The manuscript effectively highlights the relevance by emphasizing the increasing levels of PM2.5 pollution in the atmospheric environment due to the intensive development of livestock houses.
Following are my major concerns related to the manuscript.
· I noticed an unusual numbering in the abstract; typically, it should be straightforward in a single paragraph.
· While the introduction mentions the investigation of the effects of PM2.5 on the pulmonary microbiota and pulmonary inflammation in broilers, it would be helpful to explicitly state the study's specific objectives. This would provide a clear roadmap for readers and help them understand the scope of the research.
· It would be beneficial to elaborate on why broilers were chosen as the subject of the study, such as their susceptibility to respiratory diseases or their economic importance in the livestock industry.
· The author should highlight the research gap and the novelty of the present study in the introduction. How does this study contribute to the existing body of knowledge? What are the unique aspects of studying PM2.5's effects on the pulmonary microbiota in broilers? Providing this information would enhance the significance and originality of the research.
· The introduction could benefit from improved structuring and flow. Consider organizing the information into paragraphs with clear topic sentences to enhance readability and logical progression of ideas. Additionally, transitioning phrases or sentences can ensure a smooth flow between concepts and ideas.
The author needs to briefly mention the specific source or origin of PM2.5 and any relevant characteristics or properties of the particles.
· The author should provide more specific details regarding the housing conditions, including temperature, humidity, and lighting, as these factors can potentially impact the experimental outcomes.
· The author should clarify the rationale for selecting a PM2.5 concentration of 1000 μg/m3 and justify the exposure duration of 2 hours per day. Additionally, details on how the PM concentrations were monitored in real-time using the IDG100-TSP monitor would enhance the transparency of the experimental setup.
· It would be helpful to specify the specific regions or sites of the sampled lung tissues and the number of replicates used for each analysis.
· What does a significant difference (R = 0.7593, P = 0.023) between the CON and PM groups indicate regarding the microbial community composition?
· How do the alterations in the abundance of specific phyla and genera relate to pulmonary health or inflammation?
N/A
Reviewer 2 Report
An exciting and essential study looking at the effect of fine particulate matter pollution on the pulmonary microbiota of broiler chickens. Fine particulate matter is an increasing concern for human and animal health, and it is crucial to study the effects of PM on living systems and also on microbiota since more and more evidence shows that microbiota (gut, skin and pulmonary, etc.) has a distinctive role in the physiology and health of living systems.
Line 12: Rewrite the sentence starting with "And"
line 13: grammatical error
line 28: The final sentences should be reworded for better flow and ease of reading
Line 58: correct "researches" to "research"
Line 62: The whole paragraph should be reworded for better flow
line 80: Unclear
2.3 should be reworded for ease of reading
Line 128: should be rewritten
2.6:. Analysis of Histological sounds incomplete and should be rewritten
Line 142: "Pulmonary microbiota was measured" sounds unclear, and I would suggest using a better word choice
Results and Discussion should be reworded at places for scientific soundness and better flow.
It would be great to have the limitations and possible future research directions in the conclusion section.
It should be improved focusing on the grammar, scientific soundness and ease of understanding for the reader.
Reviewer 3 Report
The article: Fine Particulate Matter Perturbs the Pulmonary Microbiota in Broiler Chickens was reviewed
A study of great interest related to the production and health of broilers was carried out. However, although PM ranges found in poultry houses are presented in the Introduction, the rest of the problem description is associated with respiratory tract problems in humans, but diseases in birds are not mentioned. It is recommended to address this topic in the Introduction
In the Discussion, the association between problems in lung histology and cytokine secretion in broilers exposed to PM with the main respiratory tract problems that constantly occur in birds, especially viral diseases, is not described. It is recommended to address this topic in the Discussion
Extensive editing of English language is required
Round 2
Reviewer 1 Report
The author's adept handling of all raised questions within the paper has undoubtedly paved the way for its acceptance. By meticulously providing comprehensive and well-founded answers, the author has demonstrated their mastery of the subject matter and their commitment to scholarly rigor
Reviewer 3 Report
I have no more comments. I recommend publication of the article.